# An intranasal adjuvanted, recombinant influenza A/H5 vaccine primes against diverse H5N1 clades: a phase I trial

Meagan E. Deming [1,6], Franklin R. Toapanta [1,6], Marcela Pasetti[1], Hana Golding[2], Surender Khurana [2], Tarek Hamouda[3], Ali Fattom[3], Yuanyuan Liang[1,4], Sharon M. Tennant[1], Megan F. McGilvray[1], Paula J. Bernal [1], Jennifer J. Oshinsky [1], Shrimati Datta[1], Jasnehta Permala Booth[1], Lynda Coughlan [1,5], Kathleen M. Neuzil[1], Chad D. Costley[3], Karen L. Kotloff [1], Marcelo B. Sztein[1] & Justin R. Ortiz [1] ✉ On behalf of the rH5 Writing Group*

Mucosal influenza vaccines may provide improved protection against infection and transmission, but their development is hindered by absence of immune correlates of protection. Here, we report a randomized, controlled phase I trial of a recombinant influenza A/H5 (A/Indonesia/05/2005, clade 2.1) hemagglutinin vaccine formulated with a nanoemulsion adjuvant ($W_{80}5EC$). The vaccine is administered intranasally in two doses 28 days apart at three antigen levels. Controls receive unadjuvanted H5 or placebo. Six months later, participants receive an intramuscular boost with unadjuvanted inactivated A/H5N1 (A/Vietnam/1203/2004, clade 1) vaccine. Primary outcomes are solicited and unsolicited adverse events (AEs), laboratory safety abnormalities, medically-attended AEs, potential immune-mediated conditions, new-onset chronic conditions, and serious AEs. All vaccines are well tolerated. After the intranasal series, hemagglutination inhibition and microneutralization responses are minimal. However, adjuvanted H5 recipients show significant increases in mucosal and serum IgG/IgA, surface plasmon resonance antibody binding, memory B and CD4 T cell activity, and antibody-dependent cell-mediated cytotoxicity. Following H5N1 boost, participants mount robust responses across measurements and have microneutralization responses against diverse H5N1 clades (including circulating clade 2.3.4.4b). Findings demonstrate successful mucosal priming and broad cross-clade responses. This intranasal vaccine supports further exploration of mucosal immune biomarkers and may accelerate development of intranasal influenza vaccines. **ClinicalTrials.gov registration:** NCT05397119

Influenza viruses cause significant health and economic harm through seasonal epidemics. Additionally, animal influenza viruses threaten food supplies and occasionally cross species barriers, causing disease and outbreaks in humans. The risk of emerging zoonotic influenza viruses with pandemic potential is exemplified by the extensive spread of clade 2.3.4.4b H5N1 avian influenza viruses in poultry and livestock, with sporadic cases of human illness[1].

Current intramuscular influenza vaccines induce strain-specific systemic immune responses to the major surface glycoprotein, hemagglutinin (HA). These responses effectively prevent symptomatic

A full list of affiliations appears at the end of the paper. *A list of authors and their affiliations appears at the end of the paper.
✉e-mail: jortiz@som.umaryland.edu

illness when vaccines are well-matched to circulating strains but may be less effective at preventing infection[2,3]. In contrast, mucosal vaccines, which stimulate immune responses at the site of infection, may provide superior protection against both viral shedding and transmission of influenza[3]. Recognizing this potential, public health organizations have advocated for improved mucosal vaccines, particularly those capable of broadening immunity against diverse influenza viruses[4,5].

For traditional intramuscular influenza vaccines, a hemagglutination inhibition (HAI) titer ≥40 is considered an immune correlate of protection[6] and serves as a standard for licensure of seasonal and pandemic influenza vaccines[7,8]. In contrast, mucosal influenza vaccine development faces challenges due to the lack of established immune correlates of protection[9]. Only one mucosal influenza vaccine, a live-attenuated influenza vaccine (LAIV), is licensed in the United States[10]. This vaccine often does not meet the regulatory HAI standard[11], and its clinical development required extensive field trials to demonstrate efficacy[11]. Developing mucosal vaccines for emerging pathogens like avian influenza faces even greater challenges. For instance, a 2007

clinical trial of an H5N1 LAIV failed to elicit HAI responses[12]; however, participants who later received an intramuscular H5N1 boost exhibited significant immune recall[13], a pattern observed with other avian influenza LAIVs followed by intramuscular boosts[14–16]. For mucosal avian influenza vaccines, defining immune correlates of protection would be valuable in advancing products through clinical development.

To address these challenges, we conducted a phase I randomized controlled trial of an adjuvanted intranasal influenza A/H5 subtype vaccine. We assessed product safety and performed extensive immunologic analyses to explore markers of mucosal vaccine immune priming.

## Results

### Clinical trial design and participants

There were five vaccine groups. Three groups received clade 2.1 influenza A/H5 (A/Indonesia/05/2005) recombinant hemagglutinin glycoprotein (rH5)[17] at one of three dose levels (25, 50, and 100 μg) combined with an oil-in-water nanoemulsion (NE) adjuvant ($W_{80}5EC$)[18,19]; one group received unadjuvanted rH5 (100 μg); and one

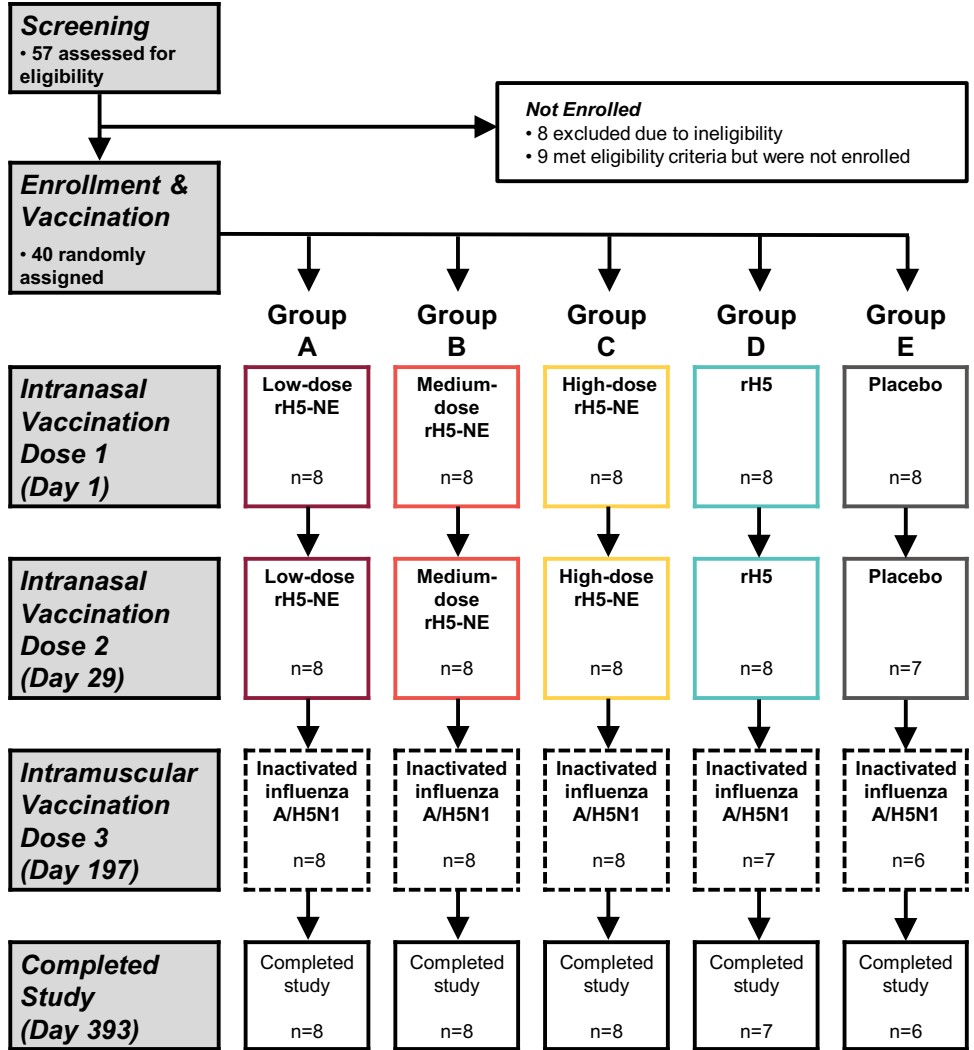

**Fig. 1 | Consort diagram.** Forty participants were enrolled and received the first vaccination. The withdrawal of one participant (Group E) before the second vaccination was not related to any safety event. A second participant (Group D) left the study area after receiving the second vaccination and could not complete the trial. A third participant (Group E) did not receive the third vaccination and was lost to follow up. He did not respond to multiple inquiries from the study team, but his emergency contact informed the study team that the participant was well. Group A (maroon), Group B (orange), Group C (yellow), Group D (cyan), and Group E (gray) are represented by these standard colors in all figures.

**Table 1 | Study design and group information**

| Vaccine allocation | Group A | Group B | Group C | Group D | Group E |
|---|---|---|---|---|---|
| Day 1 and 29 | Low-Dose rH5-NE (clade 2.1) | Medium-Dose rH5-NE (clade 2.1) | High-Dose rH5-NE (clade 2.1) | Unadjuvanted rH5 (clade 2.1) | Placebo |
| Day 197 | H5N1 IIV (clade 1) | H5N1 IIV (clade 1) | H5N1 IIV (clade 1) | H5N1 IIV (clade 1) | H5N1 IIV (clade 1) |
| n= | 8 | 8 | 8 | 8 | 8 |
| Age, median years (range) | 31.0 (19.0) | 26.5 (15.1) | 28.4 (22.0) | 28.3 (14.5) | 31.8 (15.9) |
| Sex, number (%) | | | | | |
| Female | 2 (25.0) | 3 (37.5) | 4 (50.0) | 5 (62.5) | 4 (50.0) |
| Male | 6 (75.0) | 5 (62.5) | 4 (50.0) | 3 (37.5) | 4 (50.0) |
| Ethnicity, number (%) | | | | | |
| Not Hispanic or Latino | 8 (100.0) | 6 (75.0) | 7 (87.5) | 7 (87.5) | 6 (75.0) |
| Hispanic or Latino | 0 (0.0) | 2 (25.0) | 1 (12.5) | 1 (12.5) | 2 (25.0) |
| Race, number (%) | | | | | |
| Black or African American | 2 (25.0) | 2 (25.0) | 0 (0.0) | 0 (0.0) | 3 (37.5) |
| White | 4 (50.0) | 4 (50.0) | 7 (87.5) | 8 (100.0) | 5 (62.5) |
| Other | 2 (25.0) | 2 (25.0) | 1 (12.5) | 0 (0.0) | 0 (0.0) |
| BMI, mean (standard deviation) | 26.9 (5.5) | 26.6 (5.0) | 23.2 (3.1) | 27.2 (6.0) | 23.9 (4.1) |

Data are for enrolled participants collected at the time of screening. Sex was defined as sex assigned at birth. Race and ethnicity were defined according the 1997 U.S. Office of Management and Budget (OMB) Revisions to the Standards for the Classification of Federal Data on Race and Ethnicity. Sex, race, and ethnicity were determined by participant self-report.

group received placebo. Vaccines were administered intranasally on Days 1 and 29. Six months later (Day 197), all participants received a heterologous intramuscular boost with an unadjuvanted 90 µg dose of a licensed, inactivated clade 1 influenza A/H5N1 (A/Vietnam/1203/2004) vaccine (H5N1 IIV) (Sanofi Pasteur Inc, Swiftwater, PA, 2007).

The trial was conducted from July 7, 2022, through October 12, 2023. Forty healthy adults aged 18–45 years were enrolled and randomized, with eight participants each assigned to Group A (low-dose rH5-NE), Group B (medium-dose rH5-NE), Group C (high-dose rH5-NE), Group D (unadjuvanted high-dose rH5), and Group E (placebo) (Fig. 1).

Overall, 45% of participants were women, 18% were Black or African American, and 15% were of Hispanic ethnicity. Their mean age was 30.2 years. Participant demographic and baseline information by vaccine group are in Table 1.

**Hemagglutination inhibition (HAI) responses**

Baseline immunity by HAI Geometric Mean Titer (GMT) to rH5 (clade 2.1) and H5N1 IIV (clade 1) was low among all groups (GMT ≤ 5.5), and there were no significant increases on Days 57 or 197 after intranasal vaccinations (Fig. 2 panel A and Supplementary Table 1). Twenty-eight days after H5N1 IIV was administered (on Day 225), we observed significant GMT responses among Group A and Group C compared to baseline or to Day 57 ($p < 0.05$). Geometric mean fold rise (GMFR) to rH5 (clade 2.1) at Day 225 among the rH5-NE groups was 17.4 (95% CI 4.6, 66.0) for Group A, 3.7 (95% CI 0.82, 16.4) for Group B, and 14.7 (95% CI 10.1, 21.3) for Group C, while the GMFR at Day 225 remained low for comparator groups, 1.1 (95% CI 0.9, 1.4) for Group D and 1.6 (95% CI 0.5, 5.2) for Group E. The percentage of participants with seroconversion at Day 225 to rH5 (clade 2.1) was 87.5% (95% CI 47.3, 99.7) for Group A, 37.5% (95% CI 8.5, 75.5) for Group B, 100.0% (95% CI 63.1, 100.0) for Group C, 0.0% (95% CI 0.0, 41.0) for Group D, and 16.7% (95% CI 0.0, 64.1) for Group E.

Day 225 HAI GMTs responses were significantly higher than baseline or Day 57 ($p < 0.05$) for Group A and Group C against H5N1 IIV (clade 1), with the highest titers among the rH5-NE groups (Fig. 2 panel A and Supplementary Table 1). GMFR to H5N1 IIV (clade 1) at Day 225 was 20.7 (95% CI 10.4, 41.3) for Group A, 3.1 (95% CI 0.9, 11.0) for Group B, 10.4 (95% CI 3.9, 27.5) for Group C, 7.2 (95% CI 1.1, 48.6) for Group D, and 2.2 (95% CI 0.8, 6.5) for Group E. The percentage of participants

with seroconversion at Day 225 to H5N1 IIV (clade 1) was 100.0% (95% CI 63.1, 100.0) for Group A, 37.5% (95% CI 8.5, 75.5) for Group B, 87.5% (95% CI 47.3, 99.7) for Group C, 57.1% (95% CI 18.4, 90.1) for Group D, and 33.3% (95% CI 4.3, 77.7) for Group E.

In a post hoc analysis, we conducted HAI to rH5 (clade 2.1) and H5N1 IIV (clade 1) at Day 204, seven days after the intramuscular H5N1 IIV administration (Fig. 2 panel A and Supplementary Table 1). The GMFR results at Day 204 were similar to the Day 225 values for both H5N1 clades, indicating a recall response rather than a primary immune response to the intramuscular vaccination.

**Microneutralization (MN) responses**

Similar to HAI results, we measured no increases in MN titers on Days 57 or 197 to either A/Indonesia/05/2005 (clade 2.1) or H5N1 A/Vietnam/1203/2004 (clade 1). However, significant increases in MN titers were measured at Day 225 compared to baseline and to Day 57 among the rH5-NE groups to both strains (Fig. 2 panel B and Supplementary Table 2). MN GMFR to A/Indonesia/05/2005 (clade 2.1) at Day 225 was 20.7 (95% CI 1.0, 106.7) for Group A, 14.7 (95% CI 5.7, 38.0) for Group B, 7.3 (95% CI 3.6, 15.1) for Group C, 1.1 (95% CI 0.9, 1.4) for Group D, and 1.0 (95% CI 1.0, 1.0) for Group E. The percentage of participants with seroconversion at Day 225 to A/Indonesia/05/2005 (clade 2.1) was 75% (95% CI 34.9, 96.8) for Group A, 87.5% (95% CI 47.3, 99.7) for Group B, 87.5% (95% CI 47.3, 99.7) for Group C, 0.0% (95% CI 0.0, 41.0) for Group D, and 0.0% (95% CI 0.0, 45.9) for Group E.

MN GMFR to H5N1 A/Vietnam/1203/2004 (clade 1) at Day 225 was 17.5 (95% CI 3.7, 83.2) for Group A, 6.2 (95% CI 2.6, 14.8) for Group B, 4.8 (95% CI 2.3, 10.0) for Group C, 3.6 (95% CI 0.7, 17.8) for Group D, and 1.3 (95% CI 0.9, 1.8) for Group E (Fig. 2 panel B and Supplementary Table 2). The percentage of participants with seroconversion at Day 225 to H5N1 A/Vietnam/1203/2004 (clade 1) was 87.5% (95% CI 47.3, 99.7) for Group A, 75% (95% CI 34.9, 96.8) for Group B, 75% (95% CI 34.9, 96.8) for Group C, 42.9% (95% CI 9.9, 81.6) for Group D, and 0.0% (95% CI 0.0, 45.9) for Group E.

In a post hoc analysis, we assessed the breadth of serum MN at Day 225 against a panel of additional H5N1 viruses, clade 2.2, clade 2.2.1, clade 2.3.4, and clade 2.3.4.4b (Fig. 3 and Supplementary Table 2). We only performed the MN assays against the panel of heterologous H5N1 strains if participants had a measurable MN titer against clade 2.1

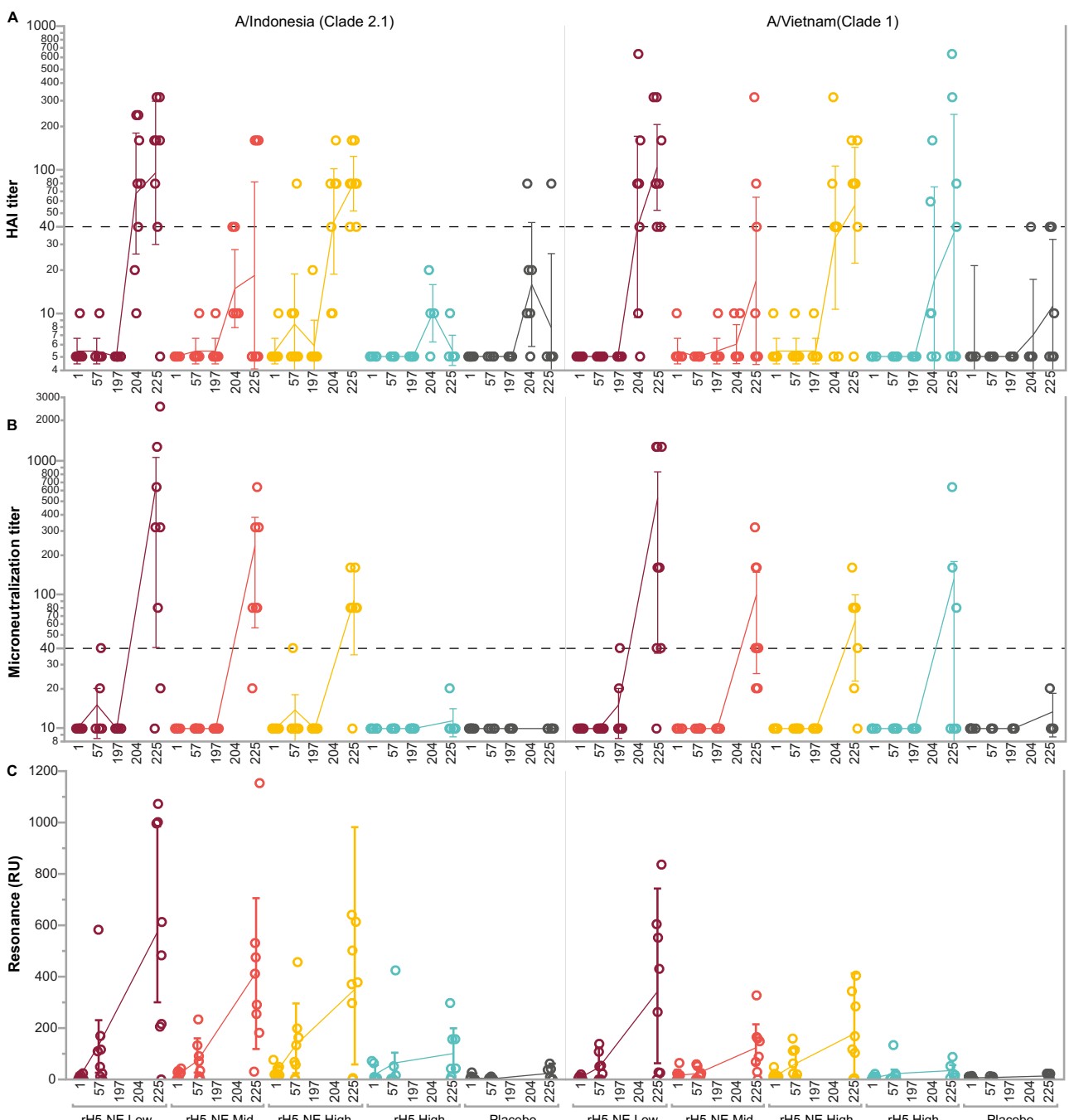

**Fig. 2 | Hemagglutination inhibition, microneutralization, and surface plasmon resonance by vaccine strain and group.** Antibody responses to influenza A/H5N1 A/Indonesia/05/2005 (clade 2.1) and A/Vietnam/1203/2004 (clade 1) at baseline, Day 57 (28 days post-second intranasal vaccination), Day 197 (immediately before intramuscular H5N1 IIV boost), and Day 225 (28 days post intramuscular H5N1 IIV boost). Serum HAI (**A**) and microneutralization (**B**) titers are shown as individual results for each participant (n = 8 per group, except as per the consort diagram in Fig. 1), with lines connecting the geometric mean titers at each time-point with 95% confidence intervals. **C** Individual Surface Plasmon Resonance (SPR)

responses with connecting mean and standard deviations. HAI assays were also conducted with specimens from Day 204 (seven days post intramuscular H5N1 IIV boost, with samples available from n = 8 participants in group A and C, n = 7 in group B, and n = 6 in groups D, E). For the HAI assay, sera that were negative at the initial dilution were assigned a titer of 5. For the MN assay, sera that were negative at the initial dilution were assigned a titer of 10. Dotted lines in (**A** and **B**) show the 1:40 dilution. Low-dose (Group A), medium-dose (Group B), and high-dose (Group C) of rH5-NE are shown in maroon, orange, and yellow. Controls, including the unadjuvanted rH5 (Group D) and placebo (Group E), are shown in cyan and gray.

and clade 1 after unblinding. In earlier studies, no background responses against this panel of H5N1 viruses were found in healthy U.S. adults[20]. rH5-NE containing vaccines elicited seroconversion against clade 2.2 (50.0%–62.5%), clade 2.2.1 (50.0%–62.5%), clade 2.3.4 (75.0%–87.5%), and clade 2.3.4.4b (75.0%–87.5%) viruses, in the reverse dose response relationship observed with MN assays against clade 2.1

and clade 1 strains. Group D seroconversion against the panel of viruses was generally lower than the rH5-NE groups.

## Serum immunoglobulin responses

We measured rH5 (clade 2.1)-specific serum IgG and IgA and H5 stalk-specific serum IgG by ELISA (Fig. 4 and Supplementary Table 3). At

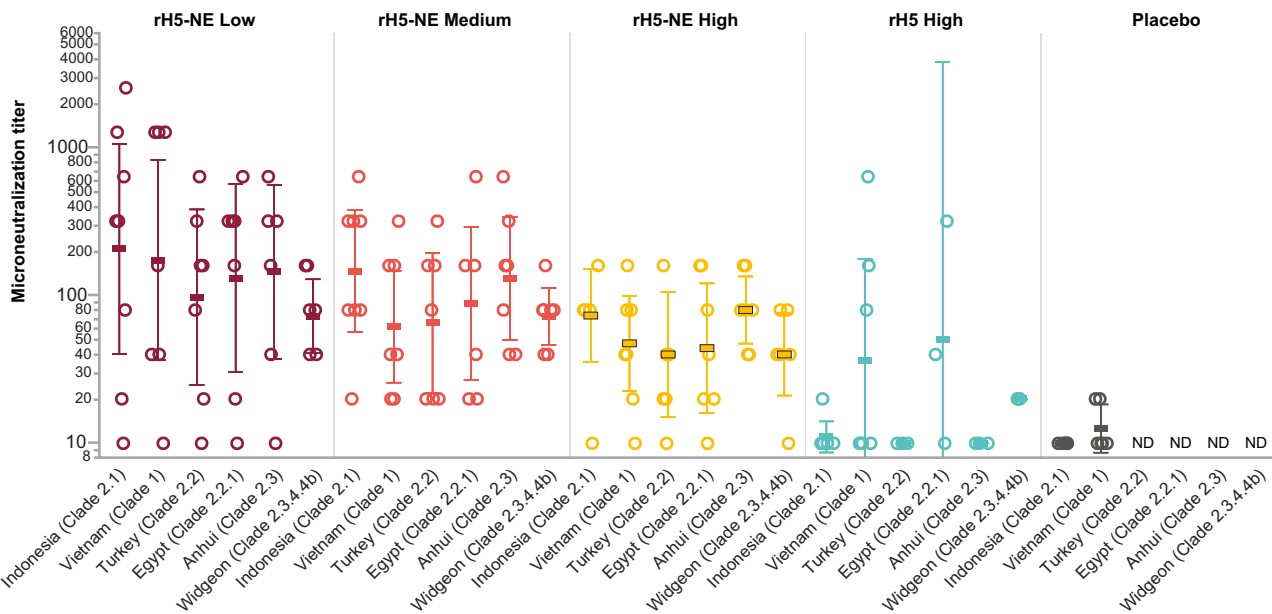

**Fig. 3 | Microneutralization on Day 225 by A/H5N1 virus panel and group.** Sera from Day 225 (28 days post intramuscular H5N1 IIV boost) were tested by MN assay using a panel of the following viruses: A/Indonesia/5/2005 (clade 2.1), A/Vietnam/1194/2004 (clade 1), A/Anhui/1/2000 (clade 2.3.4), A/Egypt/3072/2010 (clade 2.2.1), A/Turkey/15/2006 (clade 2.2), and A/Wigeon/sc/22/2021 (clade 2.3.4.4b). Sera that were negative at the initial dilution were assigned a titer of 10. Values shown as individual results for each participant, with geometric mean titers indicated by horizontal bar with 95% confidence intervals. Samples from the placebo group were not tested against the broader panel due to low MN titers against the vaccine strains (ND, not done). Low-dose (Group A), medium-dose (Group B), and high-dose (Group C) of rH5-NE are shown in maroon, orange, and yellow. Controls, including the unadjuvanted rH5 (Group D) and placebo (Group E), are shown in cyan and gray.

baseline, IgA and IgG levels to rH5 were low, while participants had high titers of anti-stalk IgG. Unlike with MN or HAI assays, significant GMT increases ($p < 0.05$) from baseline were observed for most immunoglobulins measured at Days 57 and 197 in the rH5-NE groups. Significant GMT increases were measured from Day 197 to Day 225 in all groups ($p < 0.05$) for IgG and IgA, though significant boost responses at this timepoint for H5 stalk-specific serum IgG were only measured in Group C and Group E ($p < 0.05$).

### Surface plasmon resonance (SPR) binding
We assessed the quality of humoral immune response using SPR-based real-time kinetics assay against both H5N1 clade 2.1 and clade 1 (Fig. 2 panel C and Supplementary Table 4). Antibody binding was significantly higher in the rH5-NE groups than in controls on Day 57 ($p = 0.0058$). After intramuscular H5N1 IIV, antibody binding increased in each group, with values significantly higher in the rH5-NE groups than in controls at Day 225 ($p = 0.0003$). Similar patterns were seen with SPR against H5N1 clade 2.1 and against H5N1 clade 1 rHAs, though responses were consistently higher against the rH5 vaccine clade 2.1.

### Antibody-dependent cell-mediated cytotoxicity (ADCC) responses
ADCC was assessed against rH5 clade 2.1. We defined ADCC seroconversion as ≥fourfold increase from baseline. Note that in this report, antibody seroconversion is referred to as seroconversion, while ADCC seroconversion is referred to as ADCC seroconversion. We measured ADCC seroconversion in all rH5-NE groups at Days 57 and 225 ($p < 0.05$), while comparator groups developed ADCC responses only at Day 225 (Fig. 5 panel A and Supplementary Table 5). The percentage of participants with ADCC seroconversion at Day 57 was 50% (95% CI 11.8, 88.2) for Group A, 57% (95% CI 18.4, 90.1) for Group B, and 75% (95% CI 34.9, 96.8) for Group C. ADCC seroconversion at Day 225 was 100% (95% CI 54.1, 100) for Group A, 86% (95% CI 42.1, 99.6) for Group B, 75% (95% CI 34.9, 96.8) for Group C, 33% (95% CI 4.3, 77.7) of Group D, and 50% (95% CI 11.8, 88.2) for Group E.

### Mucosal immune responses
Nasal wash rH5-specific IgG normalized by total IgG was low at baseline for all groups (Fig. 4 panel B and Supplementary Table 6). Significant increases within groups were noted at Days 43, 57, and 197 compared to baseline for all rH5-NE groups ($p < 0.05$). At Day 225, significantly increased nasal IgG responses were measured for all groups compared to Day 197 ($p < 0.05$), with the highest responses in the rH5-NE groups.

Baseline levels of nasal rH5-specific IgA, normalized by total IgA, were low across all groups. However, significant increases were seen in Groups A and B on Days 43, 57, 197, and 225 compared to baseline ($p < 0.05$), but not for Groups C, D, or E (Fig. 4 panel B and Supplementary Table 6). A significant boost response in nasal IgA levels was seen on Day 225 compared to Day 197 in Groups A and B ($p < 0.05$).

### Memory B cells
We assessed memory B cells with the potential to produce antibodies against rH5 (clade 2.1) in polyclonally expanded PBMC. Analysis of rH5-NE groups showed significant increases in vaccine-specific antibody-secreting cells (ASC) on Day 197 compared to baseline. Only Group C (high-dose rH5-NE) had a significant increase on Day 57 compared to baseline. rH5-NE groups B and C had significant increases in vaccine-specific ASCs after intramuscular H5N1 IIV on Days 57 and 197 compared to baseline (Fig. 5 panel B). The unadjuvanted rH5 group and the placebo group did not exhibit significant increases in H5-specific ASC from baseline at any subsequent timepoint.

### T cell immunity
Activated (CD69+) memory CD4 T cells (CD4+, excluding CD45RA + CD62L+ cells) were assessed for their ability to produce cytokines or upregulate additional activation markers (e.g., CD154, CD137) after ex-vivo stimulation with rH5 (clade 2.1) (Supplementary Fig. 1 **panel A**) and a H5 peptide pool (clade 1) (Fig. 5).

Peptide stimulations increased IL-2 expression significantly at Days 57, 197, and 225 compared to baseline among Groups A and B. Group C increased expression of this cytokine only after the

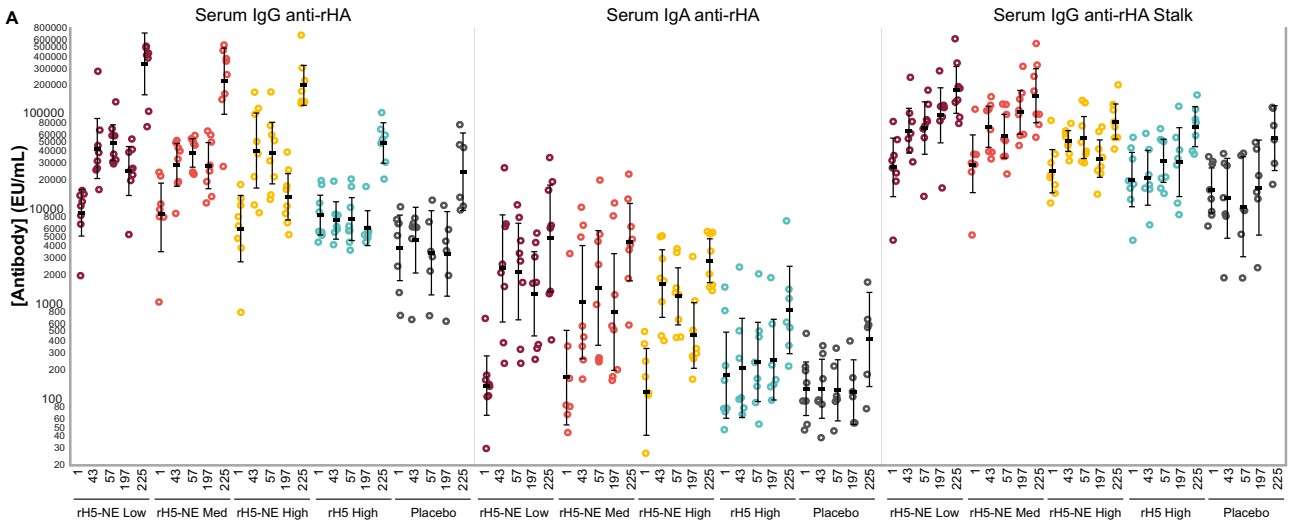

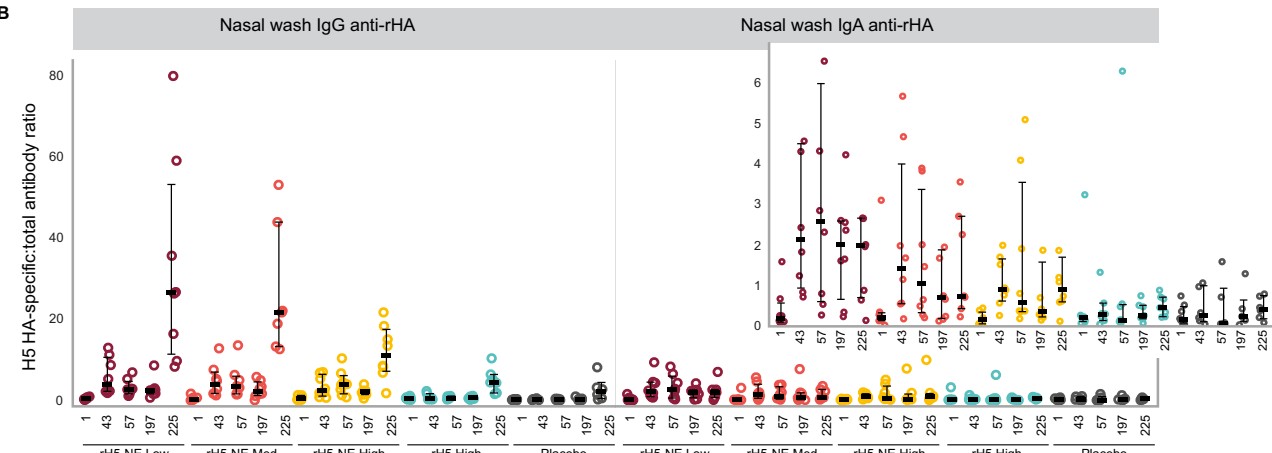

**Fig. 4 | H5N1 clade 2.1 serum and nasal wash binding antibodies by group.**
**A** shows the H5 A/Indonesia (Clade 2.1)-specific serum IgG and IgA responses, as well as H5 stalk-specific IgG responses (EU/mL); each dot represents one individual, with summary values shown as geometric mean concentration with 95% confidence intervals. **B** shows individual nasal wash responses (IgG and IgA) as well as the median and interquartile range of the ratio of H5-specific IgG or IgA (EU/µg) to total

IgG or IgA at each timepoint per group. Inset shows nasal wash responses on an extended *y*-axis. Low-dose (Group A), medium-dose (Group B), and high-dose (Group C) of rH5-NE are shown in maroon, orange, and yellow. Controls, including the unadjuvanted rH5 (Group D) and placebo (Group E), are shown in cyan and gray.

intramuscular H5N1 IIV (Day 225) (Fig. 5**panel C**). IFN-γ was upregulated by all rH5-NE groups at Day 225. Only Group A had increased expression of this cytokine prior to intramuscular H5N1 IIV (Day 197) (Fig. 5 panel D). Memory CD4 T cells stimulated with the rH5 (clade 2.1) had IL-2 and IFN-γ responses of lower magnitude than those identified in cells stimulated with the H5 peptide pool (clade 1), but the trends were similar (Supplementary Fig. 1**panels C-D)**. The comparator groups did not exhibit significant increases from baseline to Day 225 in memory CD4 T cells producing cytokines (Fig. 5 panels C and D), or in activation or degranulation markers.

Since memory CD4 T cells from rH5-NE groups stimulated with the clade 1 peptide pool had significant expression of IL-2 and IFN-γ, we assessed whether these cells had multifunctional activity, defined as production of more than one cytokine by the same cell. First, we performed these assessments in peptide-stimulated cells pooled from all rH5-NE groups (Fig. 5 panel E). We identified a significant increase in the frequency of multifunctional cells (IL-2+, IFN-γ+) on Days 57 and 197 compared to baseline. After intramuscular H5N1 IIV vaccination, the frequency of these cells increased significantly. Notably, IL-2+ single functional cells had the highest frequency at every timepoint after intranasal and intramuscular vaccination (Fig. 5 panel E).

Intramuscular H5N1 IIV increased the frequency of multifunctional memory CD4 T cells in all the rH5-NE groups (shown as percentage of mean multifunctional and single function cells), but the change was highest in Group A and Group B (Fig. 5 panel F). For example, in Group A, the frequencies of multifunctional cells changed from 8.4% (95% CI 0, 27.1) at Day 57 to 15.5% (95% CI 5.8, 90) by Day 225. We identified no significant production of cytokines or upregulation of CD107a in memory CD8 T cells upon stimulation with H5N1 clade 2.1 or clade 1.

## Safety
The rH5-NE vaccines were well tolerated. Among participants receiving the first intranasal vaccination, immediate (within 60 min) reactogenicity symptoms were common and mostly mild (Grade 1) (Fig. 6 and Supplementary Table 7). Solicited reactogenicity symptoms occurring in >5% of participants included runny nose (55.0%), postnasal drip (52.5%), stuffy nose (32.5%), sore throat (35.0%), and itchy nose (10.0%). Participants receiving rH5-NE containing vaccines experienced more local symptoms and three of four moderate-severity events (Grade 2). Immediate reactogenicity symptoms after the second intranasal dose in >5% of participants included runny nose (61.5%), postnasal drip (48.7%), sore throat (48.7%), stuffy nose (41.0%), itchy nose (5.1%),

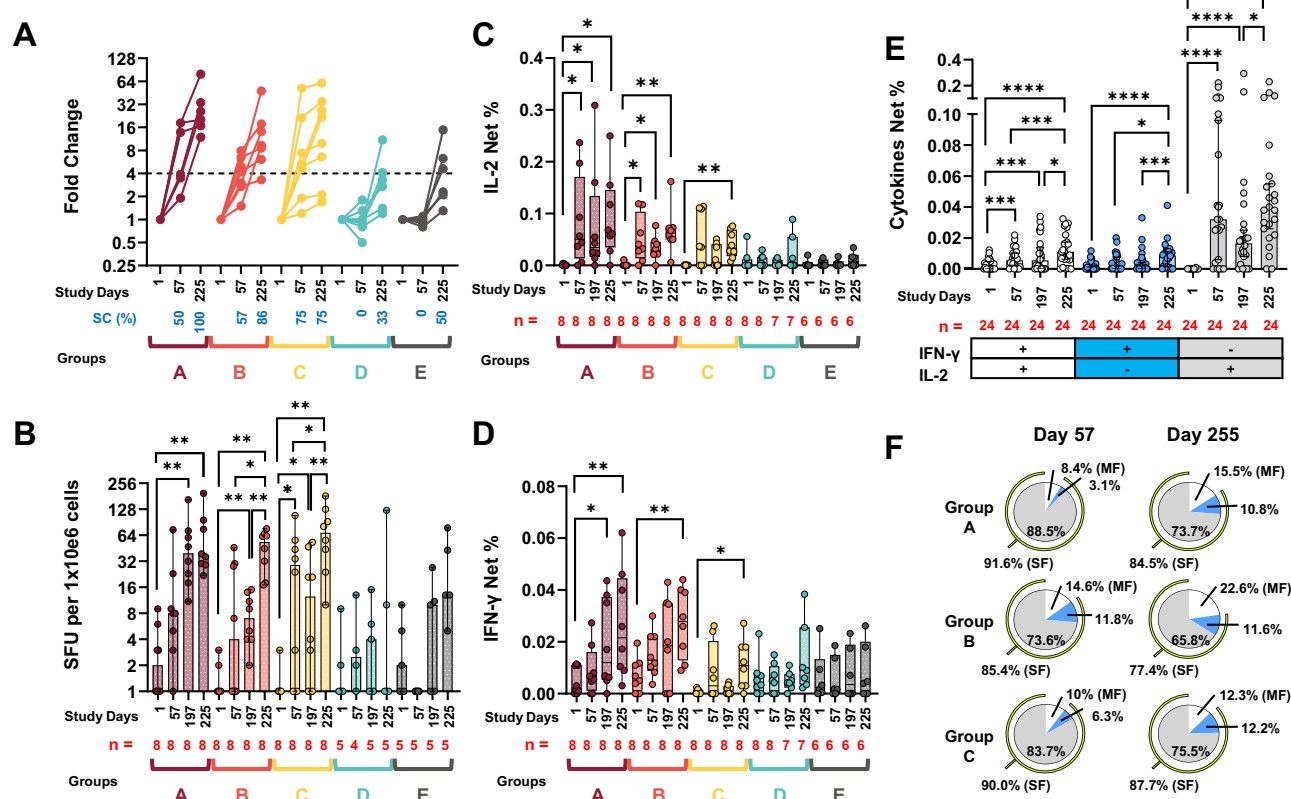

**Fig. 5 | Antibody-dependent cell-mediated cytotoxicity (ADCC), memory B cell, and memory T cell responses by group. A** shows ADCC as fold-changes over Day 1. Blue text shows the percentage of volunteers that ADCC seroconverted (SC; ≥ fourfold rise; dotted line threshold). RLU data and stats reported in Supplementary Table 5. **B** shows the frequency of memory B cells producing anti-H5 IgG antibodies (SFU per 1 × 10e6 cells). Bars indicate median ± 95% CI. In (**C** and **D**) box (25th to 75th percentiles) and whiskers (minimum and maximum data points) plots display the frequency (net %) of memory CD4 T cells producing IL-2 (**C**) and IFN-γ (**D**) upon stimulation with an H5 peptide pool (A/Vietnam/1203/2004 (clade 1)). In plots A–D volunteers vaccinated with low-dose (Group A), medium-dose (Group B), and high-dose (Group C) of rH5-NE are shown in maroon, orange, and yellow. Unadjuvanted rH5 (Group D) and placebo (Group E) are shown in cyan and gray colors,

respectively. **E** shows multifunctional (MF; IL-2+ & IFN-γ+) memory CD4 T cells (pooled Groups A–C) (white circles). IFN-γ-only and IL-2-only producing cells (Single Functional -SF- cells) are shown by the blue and grey circles, respectively. Bars indicate the median ± 95% CI. In (**A–E**), each dot represents one individual. In (**B–E**), the number of volunteers assessed at each timepoint (n) is shown in red font. **F** displays the frequency of MF and SF cells by Groups A–C at days 57 and 255. The data shown as percentage of the mean of MF and SF cells. Blue, gray, and white areas of the pie show IFN-γ-only, IL-2-only, and MF cells, respectively. Yellow semicircle shows the added percentage of SF cells (IL-2-only plus IFN-γ-only). Statistics from (**B–E**) derived from Wilcoxon signed-rank tests (2-sided) *p < 0.05, **p < 0.01, ***p < 0.005, **** p < 0.0001.

feverishness (5.1%), headache (5.1%), watery eyes (5.1%). Most symptoms occurred in rH5-NE groups, including one moderate-severity event (Grade 2). Solicited 7-day post-vaccination reactogenicity symptoms were mostly mild (Grade 1), remained more common in rH5-NE groups, and were similar to the immediate reactogenicity profiles.

Five related unsolicited adverse events within 1 h of vaccination were mild, including elevated diastolic blood pressure, toothache, sinus pain, and nasal discomfort (2 participants) (Supplementary Tables 8 and 9). Two solicited events (mild cough and postnasal drip) began during the 7-day post-vaccination period and extended beyond it, qualifying as unsolicited events. All adverse events were self-limited. There were two other related unsolicited adverse events (cough and upper-airway cough) within the 28-day post-vaccination period, and none of the related unsolicited adverse events was medically attended. No adverse events after either intranasal vaccination were of severe or higher severity.

Laboratory abnormalities within seven days post-dose 1- and 14-days post-dose 2 were mostly mild (Grade 1) (Supplementary Tables 8 and 10). Two moderate abnormalities (Grade 2) occurred after the first intranasal vaccination with no severe or higher severity events. There were no moderate or greater severity adverse events after the second intranasal vaccination.

The licensed H5N1 IIV was also well tolerated with few solicited adverse events and no related unsolicited events. Two participants had unrelated Grade 3 low hemoglobin after H5N1 IIV vaccination, assessed as due to study phlebotomy. No other moderate or greater severity events occurred after H5N1 IIV vaccination (Supplementary Tables 10 and 11).

Throughout the trial there were no potentially immune-mediated medical conditions, new onset chronic medical conditions, or serious adverse events (Supplementary Table 12).

## Discussion

In this study, we evaluated the safety and immunogenicity of an intranasal adjuvanted recombinant influenza A/H5 vaccine. Aware of the absence of an immune correlate of protection for mucosal influenza vaccines, we performed extensive assessments of cell-mediated, humoral, and mucosal immune responses to evaluate vaccine immunogenicity. As shown in prior studies of avian influenza LAIV, vaccine-induced immune responses were evident only after a subsequent intramuscular inactivated vaccine boost[13,14,16,21].

We therefore administered the intramuscular heterologous H5N1 IIV (A/Vietnam/1203/2004, clade 1) vaccine boost dose six months after the two intranasal rH5-NE doses (A/Indonesia/05/2005, clade 2.1)

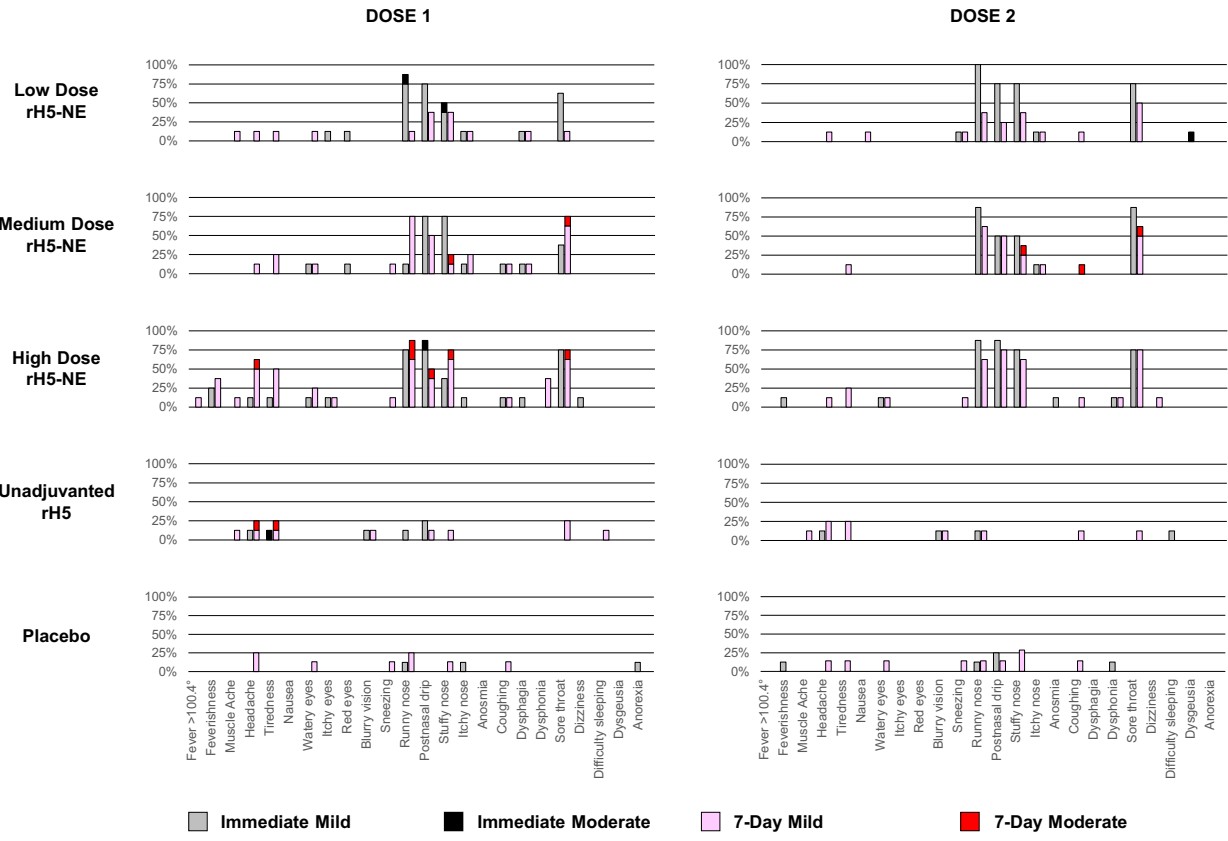

**Fig. 6 | Solicited symptoms through 7 days after each intranasal vaccination.** All solicited events were mild or moderate. No reports of nosebleed, hearing difficulty, double vision, joint pain, ear ringing, slurred speech, eye swelling, chest tightness, or wheezing. Figure colors: immediate mild (light gray), immediate moderate (black), 7-day mild (pink), 7-day moderate (red).

as an immune probe to potentially uncover vaccine priming. Like those previous studies, we observed low HAI and MN responses after the primary intranasal vaccination series but rapid, robust responses after intramuscular boost[13–16,21]. Additionally, the rH5-NE groups elicited high neutralizing antibody titers against diverse H5N1 virus clades.

The immune responses observed after the intramuscular H5N1 IIV boost indicate a strong recall response in groups that received the intranasal rH5-NE vaccines. Seroconversion rates following H5N1 IIV in the rH5-NE groups ranged from 38 to 100% for the H5N1 clade 1 virus, surpassing the 19–26% seroconversion rates after a single-dose of H5N1 IIV in a previous study[22]. The rapid HAI responses detected seven days after the H5N1 IIV boost, along with differential immune responses at 28 days post-boost in the intranasal rH5-NE groups, support a recall response of cross-reactive memory B cells rather than a primary response to the intramuscular vaccination.

We observed a reverse dose response to the rH5-NE vaccine in several immune assays, a phenomenon previously noted in adjuvanted intramuscular vaccines for avian influenza[23,24]. This finding may suggest that increased antigen doses may activate extrafollicular B cells with lower affinity, which are not targeted to the germinal centers for further affinity maturation. Such B cells are more likely to produce non-neutralizing antibodies upon boosting[25,26]. An alternative possibility is that larger antigen doses with the NE adjuvant favor the activation of CD4 regulatory T cells (Tregs) over Effector Memory CD4 T cells, with activated Tregs suppressing immune responses[27]. Some of the CD4 T cell responses that we assessed showed a similar reverse dose response pattern, though it is unclear whether the frequencies of T follicular helper cells (essential for germinal center seeding and somatic hypermutation) would also follow this pattern. Of note, ADCC-mediating antibodies followed an increasing dose response, suggesting a disassociated evolution of antibodies with different functions.

The rH5-NE vaccine was well-tolerated at all doses. Mild local nasal symptoms were common, occurred more frequently in the adjuvanted vaccines than the comparators, and were often reported within 60 min of intranasal vaccination. While rH5-NE reactogenicity was minimal in preclinical studies and our trial, we remain cautious given the history of Bell's palsy associated with an intranasal split-virus influenza vaccine adjuvanted with *E. coli* heat-labile toxin[28]. While the cause was likely related to the specific adjuvant[29], which was more immunostimulatory than the nanoemulsion used in our trial, future studies of adjuvanted intranasal vaccines should monitor for this adverse event closely.

Our study suggests that the absence of HAI or MN responses following the primary intranasal series may reflect immunological compartmentalization, with priming potentially focused on the upper respiratory tract. We saw increases in both nasal IgG and IgA levels after the intranasal vaccination series, but low magnitude nasal IgA boost after H5N1 IIV, further supporting this notion. It remains unclear whether the nasal wash immunoglobulins originated from local production or were due, at least in part, to extravasation from the vasculature. Attempts to investigate this by isolating mononuclear cells from nasal wash samples for flow cytometry were unsuccessful due to insufficient cell yields.

Despite these limitations, immunogenicity signals elicited by the primary intranasal rH5-NE series, including significant serum and mucosal IgG and IgA responses, SPR binding antibodies, and ADCC activity, warrant further investigation. ADCC, in particular, has been

historically underappreciated; however, recent studies have demonstrated that antibodies with ADCC capacity that target conserved regions of HA are cross-reactive and can be protective in adoptive transfer models[30]. Additionally, within the framework of the European Union's Innovative Medicines Initiative-funded project FLUCOP, ADCC assays have been shown to be standardized, cost-effective, and viable as an alternative immune correlate of influenza protection[31]. In our study, robust ADCC activity was detected among all rH5-NE groups following intranasal vaccination, suggesting that ADCC assays could serve as a valuable tool for evaluating mucosal vaccine priming. However, this hypothesis requires confirmation in larger, future studies.

The separate rH5-NE vaccine components have been evaluated in human trials previously. A 2010 randomized trial evaluated the nanoemulsion adjuvant combined with seasonal inactivated influenza vaccine at doses of 4–10 μg in a single intranasal dose[32]. Compared to the approved intramuscular vaccine and an intranasal placebo, 28-day HAI seroconversion ranged from 0 to 25% for intranasal groups and 60–80% for the intramuscular group. Vaccine-specific IgA levels were similar across all groups. A 2011 trial of intramuscular rH5 in a dose-ranging study (15–90 μg, with and without Alhydrogel adjuvant) found 10% seroconversion after the second dose in the best-performing group[17].

Given prior clinical experience with the nanoemulsion adjuvant and rH5 antigen, the intranasal rH5-NE formulation's performance in our study is noteworthy. In the 2010 trial, serum HAI titers were not expected to correlate with intranasal vaccine performance, and prior exposure to seasonal influenza viruses likely confounded immune response assessments. For the intramuscular rH5 study, inactivated avian influenza vaccines are known to have low immunogenicity, often requiring high antigen doses or potent adjuvants to induce measurable HAI responses[33]. These studies highlight challenges in influenza vaccine development, including unclear immunogenicity measures of success for mucosal vaccines, confounding by pre-existing immunity, and challenges advancing novel H5N1 prevention technologies.

The influenza A/H5N1 strains in the rH5 antigen (A/Indonesia/05/2005, clade 2.1) and H5N1 IIV (A/Vietnam/1203/2004, clade 1) were isolated nearly 20 years before our trial. Since then, H5N1 viruses have evolved substantially, including the emergence of clade 2.3.4.4b, which has caused widespread infections in poultry, livestock, and over 50 sporadic human cases in North America in 2024 and a recent death[34,35]. To assess cross-protective potential, we performed MN assays against a panel of H5N1 strains, including the vaccine clades and clade 2.3.4.4b. The rH5-NE groups attained the highest neutralization titers and elicited cross-protective MN responses. Similar cross-protection against H5N1 clade 2.3.4.4b has been shown with licensed adjuvanted H5N1vaccines[20]. These findings suggest the rH5-NE vaccine has potential as a preventive intervention.

Given the design of our trial, we could not directly determine whether the breadth of immunological memory induced by the two intranasal rH5-NE doses was attributable to the adjuvant or the mucosal priming. However, a 2024 study demonstrated that individuals who received two doses of an intramuscular unadjuvanted H5N1 (A/Vietnam) vaccine had significantly less cross-reactivity against heterologous H5N8 (clade 2.3.4.4b) viruses compared with those who received two doses of an adjuvanted H5N1 (A/Indonesia) vaccine[20]. In our trial, intranasal vaccination with the unadjuvanted rH5 failed to induce strong memory B cells that could be recalled by the intramuscular boost. Thus, in the absence of an adjuvanted intramuscular priming dose followed by an adjuvanted intramuscular boost, we can only hypothesize that mucosal vaccination may contribute to immune priming, but a strong adjuvant is likely critical for successful priming. The nanoemulsion adjuvant's ability to elicit strong memory immune responses at low antigen doses may reduce the required vaccine dose and help expand the available supply of avian influenza vaccines.

Our trial has notable strengths, including extensive immunological assessments of cell-mediated, humoral, and mucosal immune responses, with consistent patterns across varied readouts and signs of mucosal immune priming that warrant further investigation. The intramuscular H5N1 IIV vaccine used to probe rH5-NE priming revealed immunological priming that otherwise would not have been detected. This heterologous prime-boost approach, previously used in H5N1 trials, elicits broader neutralizing and anti-stalk antibody responses compared to homologous regimens[36–40]. The rH5-NE vaccine provides an excellent model for advancing mucosal immune priming research. It was well-tolerated, and the absence of pre-existing H5N1 immunity in the general population, simplified the interpretation of the observed immune responses. Additionally, the use of a recombinant protein vaccine avoids the safety concerns associated with H5N1 LAIV trials, which required quarantine given risks of recombination with seasonal influenza viruses[12].

Our study has several limitations. As a phase I trial, there were no pre-specified hypotheses to be tested, and the sample size was not chosen to detect group differences with sufficient statistical power. Improvements in mucosal sampling and analysis are also necessary. Variability in nasal wash volumes complicates data interpretation. Standardization of mucosal sampling could improve immune evaluation consistency and comparisons across studies. Absent advances in direct measurements of mucosal immunity, our study demonstrates the value of including an intramuscular boost to uncover mucosal priming for novel pathogens. Trial designs for evaluating mucosal vaccines against endemic respiratory diseases such as seasonal influenza and SARS-CoV-2 will need to account for preexisting immunity.

While mucosal influenza vaccines are being promoted as promising alternatives to parenteral vaccines for the prevention of infection and transmission, their pathway to licensure is hindered by the absence of defined immune correlates of protection. Our trial demonstrated that an adjuvanted rH5 influenza vaccine administered intranasally primed responses to a homologous virus and induced cross-reactive responses against a diverse panel of H5N1 viruses. Had we relied solely on standard HAI or MN measurements after the primary intranasal series, we would not have detected immunological responses. However, we had the advantage of assessing vaccine antigens against which our participants had no prior exposure, employed a heterologous intramuscular boost to reveal vaccine priming signals, and implemented extensive immunological assessments before and after the intramuscular boost to detect additional cross-reactive immune responses. These results identify potential biomarkers of successful priming for mucosal influenza vaccines, including ADCC activity and SPR antibody responses in the blood and IgG and IgA antibodies in mucosal sites. The observed immune priming responses position this vaccine as a potential dose-sparing countermeasure against H5N1 virus infection and as a valuable tool for defining biomarkers of mucosal vaccine immunogenicity. As global efforts to develop mucosal vaccines accelerate to enhance protection against respiratory virus infection and transmission, our trial underscores both the opportunities and challenges ahead.

## Methods

We conducted this study at the University of Maryland School of Medicine Center for Vaccine Development and Global Health (Baltimore, MD, USA). Healthy, non-pregnant, non-lactating adults aged 18 to 45 years were included. Participants were recruited from a volunteer database at the clinical site and through advertisements in the Baltimore/Washington, DC, region. Those meeting all eligibility criteria—including screening lab tests, negative pregnancy tests (when applicable), medical history, and physical exam—within 28 days before the first vaccination were eligible for enrollment. Individuals with chronic medical conditions or medications affecting safety or study endpoints were excluded.

Healthy adult participants were randomized into five groups: three groups received rH5-NE at different antigen levels (25, 50, and 100 μg), one group received unadjuvanted rH5 (100 μg), and one group received a formulation buffer placebo. Participants were enrolled and randomized sequentially in a stepwise, dose-escalation process using four vaccination cohorts. Each rH5-NE dose level was assessed in two sentinel participants before subsequent doses were administered.

Intranasal vaccinations were given on Days 1 and 29. On Day 197, all participants received a heterologous intramuscular boost with a 90-μg dose of licensed, inactivated H5N1 IIV. Local and systemic immune responses were measured, and vaccine safety was assessed. Participants were followed through Day 393 for safety and immunological endpoints.

The primary objective of the study was to evaluate the safety and reactogenicity of rH5-NE. Secondary objectives included assessing mucosal immune responses (IgA, IgG, and T-cell mediated immunity [T-CMI]) and humoral immune responses (serum HAI) after two doses of rH5-NE, and the safety and reactogenicity of a single booster dose of intramuscular H5N1 IIV.

### Pre-specified exploratory objectives were as follows

1. To assess humoral immune responses to rH5-NE and H5N1 IIV vaccine-specific antigens at multiple timepoints.
2. To evaluate primary T-CMI in peripheral blood to rH5-NE and H5N1 IIV vaccine-specific antigens.
3. To measure memory B-cell responses by ELISpot in peripheral blood to rH5-NE and H5N1 IIV vaccine-specific antigens.

On May 29, 2024, we modified the clinical trial protocol. Previous versions of the protocol included rH5-NE safety and mucosal immunogenicity as co-primary objectives. Given the spread of avian influenza A/H5N1 (clade 2.3.4.4b) through the U.S. agricultural sector and reports of human infections, the study team, in consultation with the funder, sponsor, and IRB, determined that there was a compelling public health interest in expediting trial analyses. Mucosal immunogenicity was demoted to a secondary objective in the protocol. After all clinical data were locked, a biostatistician completed analyses of available immunogenicity data and made results available to NIH, BARDA, the sponsor, and the investigators as aggregated, summary data by study arm and timepoint. Laboratory personnel responsible for completing any pending assays (including mucosal immunogenicity assays) remained blinded to participant vaccine allocation until the assays were completed.

### Study vaccines

The rH5-NE vaccine consisted of three components: rH5 glycoprotein antigen, 60% $W_{80}5EC$ nanoemulsion adjuvant, and a proprietary formulation buffer[18,19,41]. The antigen was a recombinant hemagglutinin (HA) glycoprotein expressed in the tobacco plant *Nicotiana benthamiana*, produced by Fraunhofer USA Center for Molecular Biotechnology (Plymouth, MI, USA), lot CMB-BDS-0100-017. It was a fusion protein containing:

1. HA from derived from A/Indonesia/05/2005 (H5N1, clade 2.1),
2. A poly-histidine tag (6xHis) to enhance purification efficiency, and
3. The endoplasmic reticulum retention signal (KDEL) for efficient protein expression.

$W_{80}5EC$ is a 60% nanoemulsion adjuvant manufactured by Blue-Willow Biologics, Inc. (Ann Arbor, MI) under GMP conditions using high shear homogenization of water, ethanol, cetylpyridinium chloride, Tween-80, and highly refined soybean oil to form an oil-in water nanoemulsion with a mean particle size of ~400 nm. Its vaccine ontology identifier is VO_0005284 in the NIAID Vaccine Adjuvant Compendium (https://vac.niaid.nih.gov/).

The final vaccine consisted of 20% $W_{80}5EC$, achieved by mixing 60% $W_{80}5EC$ with rH5 and formulation buffer on-site. Two comparator groups received intranasal vaccines: one with unadjuvanted rH5 mixed with formulation buffer and the other with formulation buffer alone. After completing the intranasal vaccine primary series, all groups received H5N1 IIV, produced by Sanofi (Swiftwater, PA, USA), lot #U2147A/UD08916. This licensed, split-virion vaccine, derived from A/Vietnam/1203/2004 (H5N1, clade 1), is approved for intramuscular administration as a two-dose series for persons aged 18 to 64 years at increased risk of exposure to the H5N1 influenza virus subtype contained in the vaccine (Sanofi Pasteur Inc, Swiftwater, PA, 2007). Manufactured in 2004–2005, it contains 90 μg HA per 1 mL dose. Stability testing conducted in August 2022 confirmed it met manufacturer stability specifications.

### Procedures

Participants provided written informed consent before any data collection or study procedures. Eligible participants were enrolled, randomized to a group, and vaccinated. The randomization list for the two intranasal vaccinations was prepared by an unmasked study statistician using block randomization for each cohort. If all participants proceeded to the second vaccination, the final vaccine allocation ratio would be 1:1:1:1:1.

For the intranasal vaccinations, participants and study staff involved in post-vaccination assessments were blinded to vaccine allocation. Due to the cloudy appearance of rH5-NE and the clear unadjuvanted or placebo formulations, the study staff administering the vaccines were unblinded. These staff members managed vaccine accountability, storage, preparation, and administration, but were not involved in subsequent study assessments. The third vaccination, using H5N1 IIV for all participants, was conducted in an open-label manner.

Intranasal vaccination was performed with a metered electronic pipette, administering ten successive 25 μL droplets onto bilateral inferior turbinates, for a total dose of 500 μL. Intramuscular H5N1 IIV was administered in the preferred deltoid, following manufacturer guidelines (Sanofi Pasteur Inc, Swiftwater, PA, 2007). After vaccination, participants were observed in the study clinic for 60 min (after intranasal vaccines) or 30 min (after H5N1 IIV). Participants had vital signs recorded, underwent examination by a study clinician (a required procedure after intranasal vaccinations and performed only if needed for H5N1 IIV), completed an immediate reactogenicity assessment, received electronic symptom diary training, and were discharged.

Participants were contacted by phone three days after each vaccination to assess adverse events (AEs), reinforce electronic diary instructions, and remind them of their next appointment. Participants returned to the clinic after seven days (for the first and third vaccinations) and 14 days (for the second vaccination) for safety assessments on Days 8, 43, and 204. At these visits, a study investigator reviewed electronic diaries, evaluated vaccine reactogenicity, and assessed other safety events. Safety hematology and chemistry lab evaluations were performed on venous blood samples collected at these visits.

An independent Safety Monitoring Committee (SMC) convened after seven days of safety data were collected for each of the first three cohorts. The SMC applied standardized rules to determine whether the study could proceed with vaccination of the remainder of participants at the same intranasal vaccine dose level and additional participants at the next intranasal vaccine dose level.

In-person safety and immunogenicity assessments were conducted on Days 57 and 225. Each visit included standardized neurologic, otorhinoscopic, and respiratory examinations. Blood samples (70 mL) for peripheral blood mononuclear cell (PBMC) isolation were collected on Days 1, 43, 57, 197, and 225. Serum samples (17 mL) were collected on Days 1, 29, 43, 57, 197, 204, and 225, and nasal wash samples (3–7 mL) were obtained at screening and on Days 8, 43, 57,

197, 204, and 225. These specimens were evaluated for vaccine immune responses. Safety phone calls were conducted on Day 90 and at trial close-out (Day 393).

Study data were collected and managed using REDCap electronic data capture tools hosted at the University of Maryland Baltimore[42,43]. REDCap (Research Electronic Data Capture) is a secure, web-based software platform designed to support data capture for research studies, providing (1) an intuitive interface for validated data capture; (2) audit trails for tracking data manipulation and export procedures; (3) automated export procedures for seamless data downloads to common statistical packages; and (4) procedures for data integration and interoperability with external sources.

## Safety assessments

We assessed vaccine reactogenicity based on the occurrence of solicited local and systemic reactions from the time of each vaccination through seven days post-vaccination. Reactogenicity assessments varied by the route of administration:

- Intranasal rH5 Vaccinations: Solicited local reactions included watery eyes, itchy eyes, red eyes, blurry vision, double vision, swelling around the eyes, sneezing, runny nose, postnasal drip, stuffy nose, itchy nose, inability to smell, bleeding from the nose, coughing, difficulty with hearing, ringing in ears, tightness in the chest, wheezing, trouble swallowing, hoarse voice, sore throat, slurred speech, dizziness, difficulty sleeping, food tasting strange, and decrease in appetite. Solicited systemic reactions included fever (> 38 °C), feverishness, joint pain, body aches/ muscular pain, headache, tiredness, and nausea.
- Intramuscular Vaccinations: Solicited local reactions included pain, tenderness, erythema/redness, and induration/swelling. Solicited systemic reactions included nausea/vomiting, diarrhea, headache, fatigue, and myalgia.

Unsolicited non-serious AEs and medically attended AEs were recorded from the time of each vaccination through 28 days post-vaccination. Serious adverse events (SAEs), new-onset chronic medical conditions (NOCMCs), and potentially immune-mediated medical conditions (PIMMCs) were monitored for 12 months following the final intranasal vaccination (through Study Day 393). Safety laboratory AEs and unsolicited AEs were assessed for severity, clinical significance, and causality (related or unrelated) by a study clinician using standardized grading criteria.

## Immunogenicity assessments

We assessed immune responses by several assays at multiple time-points throughout the study.

## Mucosal immune evaluations included

H5 clade 1-specific and total IgA and IgG by ELISA.
Humoral immune evaluations included:
- HAI for H5N1 A/Vietnam/1203/2004 (clade 1) and A/Indonesia/05/ 2005 (clade 2.1) at baseline (Day 1) and Days 57, 197, and 225. A post hoc analysis included HAI for H5N1 A/Vietnam/1203/2004 (clade 1) and A/Indonesia/05/2005 (clade 2.1) at Day 204.
- ELISAs for H5 A/Indonesia/05/2005 (clade 2.1) -specific IgG and IgA and H5 stalk IgG at baseline (Day 1) and Days 43, 57, 197, and 225.
- Post hoc analyses of viral microneutralization (MN) for H5N1 A/ Vietnam/1203/2004 (clade 1) and A/Indonesia/05/2005 (clade 2.1) at baseline (Day 1) and Days 57 and 225, as well as MN for H5N1 A/ Turkey/15/2006 (clade 2.2), A/Egypt/3072/2010 (clade 2.2.1), A/ Anhui/1/2000 (clade 2.3.4), and A/Wigeon/SC/22/2021 (Clade 2.3.4.4b) at Day 225 only.
- Post hoc analyses of surface plasmon resonance (SPR) binding for H5N1 A/Vietnam/1203/2004 (clade 1) and A/Indonesia/05/2005 (clade 2.1) rHA proteins at baseline (Day 1) and Days 57 and 225.
- Post hoc analyses of antibody-dependent cell-mediated cytotoxicity (ADCC) for A/Indonesia/05/2005 (clade 2.1) at baseline (Day 1) and Days 57 and 225.

Cellular immune evaluations included:
- T cell immune responses:

Activation markers (e.g., CD69, CD137, and CD154), cytokines/chemokines, and degranulation markers in CD4 and CD8 T cell memory subsets by flow cytometry for H5N1 clade 2.1 at baseline (Day 1) and Days 57, 197, and 225. Post hoc analyses included the same assays for H5N1 clade 1.

- Memory B cell immune responses:

H5-specific memory B cells measured by ELISpot for H5N1 clade 2.1 at baseline (Day 1) and Days 57, 197, and 225.

## Total and antigen-specific IgG and IgA ELISAs

For measurements of total IgA or IgG, microplates were coated with 0.5 μg/mL of purified anti-IgA (α-chain specific) or anti-IgG (γ-chain specific; Jackson Immunoresearch -JIR-, PA) overnight at 4 °C. For the detection of antigen-specific IgA or IgG, plates were coated with 2 μg/ mL of A/Indonesia/5/05/2005 (H5N1; clade 2.1) rH5 (Fraunhofer USA Center for Molecular Biotechnology) or with 0.5 μg/mL of headless H5 stalk- based on H5: A/Indonesia/05/05 stabilized[44] and produced in-house. After coating, plates were washed with PBS containing 0.05% Tween 20 (PBST) and blocked with PBST containing 5% nonfat dry milk (PBSTM) for 1 h at 22 °C. Nasal wash samples were diluted 1:1000 and 1:5000 in PBSTM for total IgA and total IgG measurements and 1:5 for rH5-specific IgA and IgG.

Serum samples were diluted 1:10,000 for quantification of rH5-specific IgA and IgG, and 1:2000 for H5 stalk-specific IgG. Sera were added to the plates and incubated for 2 h at 22 °C, washed with PBST, and incubated with biotinylated goat anti-human Fc-specific IgA or IgG (JIR, PA) diluted 1:5000 in PBSTM for 1 h at 22 °C.

After washing, TMB substrate was added to the wells and allowed to develop for 1–10 min at room temperature (in the dark and with shaking). The reaction was stopped by adding 100 μL/well of 1 M phosphoric acid (Sigma). Antibody concentrations were determined through interpolation of Absorbance values from standard curves. An in-house standard generated from pooled positive samples was used to calculate rH5-specific IgA and IgG and H5-IgG stalk. The in-house standard was assigned arbitrary unitages as ELISA Units (EU/ml) for each antibody specificity, which corresponded to the inverse of the dilution of standard that produces an $OD_{450}$ value = 0.5 above the blank. Total IgA or IgG standards were purchased: human IgA (Calbiochem, Temecula, CA) and human IgG (Millipore Sigma, St. Louis, MO).

## Hemagglutination inhibition (HAI) assay

The HAI assay was performed following a standardized protocol[45]. Influenza virus was propagated from a PR8-backboned A/Indonesia/ 05/05, a H5N1-E4 vaccine seed virus or A/rgVN/1203/2004, reverse-genetics engineered vaccine seed virus, both provided by the Food and Drug Administration Center for Biologics Evaluation (FDA/CBER). A solution of 0.5% horse red blood cells (RBCs) was used for improved

sensitivity to H5N1 viruses. Virus potency was confirmed by back-titration. Serum-only and RBC-only controls were included along with positive controls (a human polyclonal reference antisera to Influenza A/Indonesia/05/2005, BEI Resources NR-33668 and the WHO International Standard for antibody to influenza H5N1 virus, NIBSC 07/150 for A/Vietnam/1203/2004). All serum samples were treated with receptor-destroying enzyme prior to testing and then serially diluted in 96-well U-bottom plates. Influenza virus stocks, stored at −80 °C, were thawed immediately before use and the viral hemagglutination unit adjusted to 8 units in PBS, pH 7.4. The virus was then added to the sample-containing wells at a 1:1 ratio, and plates were incubated for 30 min at room temperature. Subsequently, 0.5% RBCs were added to each well and the plates incubated for 60 min at room temperature. Antibody titers were determined by the CypherOne HAI plate reader (InDevR, software version 4.0.0.19)[46] as well as visually assessing the plates for hemagglutination. The antibody titer was defined as the reciprocal of the highest serum dilution that contains an RBC agglutination inhibition precipitation pattern that is similar in size, well clarity, and morphology to the positive serum control.

### Antibody titer determination by microneutralization (MN) assay

Vaccine-induced antibody titers were assessed by virus MN assays[20]. MN titers assessed viral-neutralizing activity in MDCK cells based on the methods of the pandemic influenza reference laboratories of the Centers for Disease Control and Prevention (CDC), with minor modifications provided in an updated protocol issued by the CDC. MN titers were measured against H5N1 vaccine strains of A/Vietnam/1194/2004 (clade 1), A/Indonesia/5/2005 (clade 2.1), A/Anhui/1/2000 (clade 2.3.4), A/Egypt/3072/2010 (clade 2.2.1), A/Turkey/15/2006 (clade 2.2), and A/Wigeon/sc/22/2021 (clade 2.3.4.4b). These vaccine strains consist of the relevant HA and NA on the PR8 backbone and with the HA polybasic cleavage site removed. All MN assays were conducted at BSL-2.

Sera were tested at an initial dilution of 1:20, and those that were negative (< 1:20) were assigned a titer of 10. The titers represent the dilution of serum with highest dilution that completely suppressed virus replication. All sera were tested in triplicate, and the geometric mean value was used for analysis. The MN assays were performed at least twice independently for each serum sample.

### Binding antibody measurements by Surface Plasmon Resonance (SPR)

Steady-state equilibrium binding of post-H5N1 vaccinated human sera was monitored at 25 °C using a ProteOn SPR biosensor (Bio-Rad)[20,47,48]. The recombinant HA globular domain (rHA1-His$_6$) for the A/Indonesia/05/2005 (clade 2.1) or from H5N1- A/Vietnam/1203/2004 (clade 1) influenza virus strain was coupled to a GLC sensor chip with amine coupling with 1000 RU in the test flow cells. Samples of 200 µl of sera at tenfold dilutions were injected at a flow rate of 50 µl min$^{-1}$ (120-s contact time) for association, and disassociation was performed over a 600-s interval. Responses from the protein surface were corrected for the response from a mock (no coating) surface and for responses from a separate, buffer-only injection. Binding antibodies were determined from two independent SPR runs.

### Memory B cell assay

Cryopreserved PBMC were thawed[49,50] and polyclonally expanded (5–6 days)[50–53]. Expanded PBMC were harvested, counted, and seeded in quadruplicate wells (250,000 cells per well) of multi-screen plates coated with rH5 from A/Indonesia/5/2005 (clade 2.1) (Fraunhofer USA Center for Molecular Biotechnology) at 3 µg/mL. Controls included wells coated with 1) goat anti-human IgA (Total IgA control; 5 µg/mL) (JIR, PA); 2) goat anti-human IgG (Total IgG control; 5 µg/mL) (JIR, PA); or 3) 1× PBS. In Total IgG and IgA control wells, 2-fold dilutions of the cells, starting at 24,000, were seeded in duplicate[51–54]. Five hours later

the plates were washed and incubated with goat-anti-human IgA-HRP (JIR, PA) or goat-anti-human IgG-HRP (JIR, PA). IgG and IgA Spot Forming Cells (SFC) were visualized with AEC substrate. Total and antigen-specific B memory SFC were calculated as SFC/10e6 cells.

### Assessment of cytokine production by T cells

Cryopreserved PBMC were thawed and rested for 4 h (37 °C, 5% CO$_2$), then washed and partitioned into four $2 \times 10e6$ cell aliquots. Two aliquots were stimulated with 1) rH5 from A/Indonesia/5/2005 (clade 2.1) (Fraunhofer USA Center for Molecular Biotechnology) at 3 µg/mL; and 2) a H5 HA peptide pool from A/Vietnam/1203/2004 (clade 1) at 2 µg/mL of each peptide. The peptide pool consisted of a 93-peptide array (12- or 17-mers, with 11aa overlap) from BEI Bioresources (NR-18974). The other two aliquots served as negative (media) and positive (Staphylococcal enterotoxin B -SEB-; 10 mg/mL) controls. All samples received anti-CD107a and anti-CD28/CD49d co-stimulatory antibodies (BDB, USA). Two hours later, Brefeldin A and Monensin were added and incubated overnight (16 h, at 37 °C, 5% CO$_2$). Next day the cells were stained for flow-cytometry[49,55–58]. Briefly, cells were stained for viability (fixable yellow staining dye; 20 min; RT) and then with a surface antibody cocktail (30 min, RT) that included: CD62L, CD4, CD19, CD56, CD3, CD8, and CD45RA. The cells were then fixed, permeabilized and stained with an intracellular cocktail (30 min, RT) including the next antibodies to: CD69, IFN, IL-17A, TNF-α, CD154, IL-2, and CD137. Supplementary Table 13 lists all antibodies used, including fluorochromes, clones, provider, and lot numbers. Cells were then fixed (1% PFA) and the samples collected in a custom LSRII flow cytometer (BD, USA). FCS files were analyzed using FlowJo (Tree Star, San Francisco, USA).

### Antibody-dependent cell-mediated cytotoxicity (ADCC) assay

rH5 from A/Indonesia/5/2005 (clade 2.1) (Fraunhofer USA Center for Molecular Biotechnology) was biotinylated (Abcam, USA) and then used to coat polystyrene-streptavidin beads (SA-beads; Spherotech, USA). For the assay, each reaction used 5 µL of SA-beads coated with ~250 ng of rH5. Plasma from vaccinated volunteers (25 µL per reaction) was incubated with rH5-SA-beads (overnight, 4 °C), the beads were subsequently washed with assay media (cRPMI with 10% ultra-low IgG FBS) and added into a 96-well plate (50 µL per well). 50 µL of $1.5 \times 10e6$/mL a reporter Jurkat cell suspension were added to the beads and incubated for 5 h (37 °C; 5% CO$_2$). This Jurkat cell line expresses the firefly luciferase gene under the control of NFAT response elements and constitutively expressing CD16a (V158) (BPS Bioscience, USA). The luciferase signal was detected by a luminometer after cell lysis and the addition of luciferin substrate (BPS Bioscience, USA). The assay was performed in triplicate wells.

### Interim immunogenicity analysis

A pre-defined interim analysis of hemagglutination inhibition data were undertaken after the last in-person study visit. The study biostatistician made results available as aggregated, summary data by blinded study arm and timepoint to the study team, funder, sponsor, and BARDA. With the exception of the biostatistician, investigators remained blinded to vaccine allocation until database lock.

### Statistical analysis

There were no pre-specified hypothesis tests for this phase I trial. The sample size of 40 with 8 persons per vaccine group was chosen without the intention that group differences would be detected with a sufficient power, but was consistent with FDA guidance[59].

Safety endpoints were evaluated using the safety dataset, all participants who received at least one vaccination. For data presented separately for each vaccination, the number of participants is the number of subjects who received the corresponding vaccination (i.e.,

dose 1, 2, or 3) with non-missing data. Safety endpoints were tabulated by severity and relatedness.

The analysis population includes all participants who received study vaccination and for whom data were available at any particular timepoint. Outcome data were analyzed by vaccine group. Demographic and medical data were summarized using descriptive statistics. Mucosal IgG and IgA were summarized by ELISA units, fold rise, and whether ≥fourfold rise was achieved. Given the low cell yields, mucosal T cell responses were not analyzed. HAI, MN, and ELISA data were summarized by GMT, GMFR, and proportion of seroconversion. We defined antibody seroconversion as either a pre-vaccination titer <10 and a post-vaccination titer ≥40 or a pre-vaccination titer ≥10 and a minimum of fourfold rise in post-vaccination titer. In post hoc analyses, when there was no baseline titer information, we defined antibody seroconversion as a post-vaccination titer ≥40, and we defined ADCC seroconversion as ≥fourfold rise over baseline ADCC analysis. In this report, the term seroconversion refers specifically to antibody seroconversion, whereas ADCC seroconversion is used to denote seroconversion based on antibody-dependent cell-mediated cytotoxicity (ADCC) activity.

For binary outcomes such as seroconversion (Yes/No), proportion was calculated for each group with the corresponding exact 95% confidence intervals using the Clopper-Pearson exact method. At each time point of interest, the difference in proportions between the vaccine groups was compared using Fisher's exact test or Chi-square test as appropriate, and the difference in continuous outcomes of interest between the groups were compared using Kruskal–Wallis *H*-test. Pairwise comparisons were conducted when needed. Wilcoxon signed-rank test was used to compare differences in antibody titers between time points within each vaccine group. No adjustments were made for multiple comparisons. All statistical tests were two-sided, and all statistical analyses were performed using Stata/SE version 18 (Stata Corp, College Station, USA), and all figures were generated using Microsoft Office software, GraphPad Prism (v.10) (Boston, MA), and JMP (v.18, SAS Institute, Cary, NC). Safety assessments were performed using SAS version 9.4 (Copyright ® 2016 SAS Institute Inc).

### Regulatory and ethics

The protocol and informed consent forms were reviewed and approved by the University of Maryland, Baltimore Institutional Review Board (HP-00086502) and the study was registered at Clinicaltrials.gov (NCT05397119). The full trial protocol and statistical analysis plans have been submitted to Clinicaltrials.gov for public access. The trial protocol is also available in the **Supplementary Information**.

### Role of the funding source

Representatives from BlueWillow Biologics, Inc. contributed to study design, interpretation of study data, and writing of the manuscript. Representatives from NIAID and BARDA provided financial support or vaccine antigen for this study and offered advice on immunogenicity and safety assessments, but had no role in the design, conduct, analysis, interpretation, or writing of the manuscript. Justin R. Ortiz had full access to all the data in the study, takes responsibility for the integrity of the data and the accuracy of the data analysis, and had final responsibility for the decision to submit for publication.

### Reporting summary

Further information on research design is available in the Nature Portfolio Reporting Summary linked to this article.

### Data availability

The data supporting the findings are summarized in the manuscript. The data are stored in a controlled-access system at the University of Maryland School of Medicine. Access to the full dataset is restricted by institutional policies and the need to protect participant privacy. As the study is no longer funded, substantial resources would be required to prepare a fully annotated, de-identified dataset. Nevertheless, de-identified data can be provided for research purposes within 3 months of a request and will remain available for up to one year after access is granted. Requests should be submitted to the corresponding author and will involve collaborative engagement with the study team to ensure accurate interpretation and adherence to ethical requirements. Reagents will also be made available upon reasonable request.

### Code availability

The study did not use custom code or mathematical algorithms to generate results.

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

## Acknowledgements

We thank our study volunteers, and Center for Vaccine Development and Global Health administrative, clinical, laboratory, recruiting, and regulatory staff. We thank the members of our Safety Monitoring Committee, Andrea Berry, Joel Chua, and Kirsten Lyke. We owe many people thanks for technical advice, including Kanta Subbarao and colleagues from BARDA (Vittoria Cioce, Ruben Donis, Gary Horwith, Kristina Lu, Christine Oshansky, and Bai Yeh), National Institutes of Allergy and Infectious Diseases (Tatiana Beresnev, Brooke Bozick, Sonja Crandon, Mohamed Elsafy, Sonnie Kim, Francisco Leyva, and Melinda Tibbals), Fraunhofer USA Center for Molecular Biotechnology (Brian Green, Stephen Streatfield, and Yoko Shoji), and BlueWillow Biologics, Inc. (George Arida, Vira Bitko, and Shyamala Ganesan, Francie Kivel, and Kevin Trabbic). We thank Dr. Hang Xie from the FDA for providing reverse-genetics engineered H5N1 virus used in the HAI assays. We thank Marion Major and Marina Zaitseva, Division of Viral Products, CBER, FDA for their review of the manuscript. We acknowledge the Flow Cytometry and Mass Cytometry Core Facility of the University of Maryland School of Medicine Center for Innovative Biomedical Resources, Baltimore, Maryland, where all the flow cytometry experiments were performed. The following reagent was obtained through BEI Resources, NIAID, NIH: Peptide Array, Influenza Virus A/Vietnam/1203/2004 (H5N1) Hemagglutinin Protein, NR-18974. The study was funded by a grant from the National Institutes of Allergy and Infectious Diseases (U01AI148081). Anti-stalk assays were supported in part by funding awarded to Lynda Coughlan (R01AI148369). Mucosal IgG and IgA, and B-cell immune responses were performed under a contract between BlueWillow Biologics, Inc. and the University of Maryland, Baltimore. Meagan E. Deming was supported by an NIH-NIAID Mentored Clinical Scientist Career Development Award (K08AI70950). Influenza A rH5-NE vaccine was donated by BlueWillow Biologics, Inc. Influenza Virus Vaccine, H5N1 was provided by the U.S. Department of Health and Human Services; Administration for Strategic Preparedness and Response; Biomedical Advanced Research and Development Authority, under Contract HHSO100200600021C. The contract and federal funding are not an endorsement of the study results, product, or company. Influenza Virus Vaccine, H5N1 was manufactured by Sanofi. The manufacturer was provided the opportunity to review a preliminary version of this manuscript for factual accuracy, but the authors are solely responsible for final content and interpretation. The funders had no role in study design, data collection and analysis, decision to publish, or preparation of the manuscript. The content of this publication and federal funding do not necessarily reflect the views or policies of the U.S. Department of Health and Human Services, nor does mention of trade names, commercial products, or organizations imply endorsement by the U.S. Government. Opinions, interpretations, conclusions, and recommendations are those of the author and are not necessarily endorsed by the U.S. Government.

## Author contributions

Conceptualization (A.F., F.R.T., J.R.O., K.L.K., K.M.N., M.B.S., T.H., Y.L., D.M.S.). Methodology/Data Acquisition (F.R.T., H.G., J.P.B., J.R.O., L.C., M.P., M.B.S., M.E.D., M.F.M., P.J.B., J.J.O., S.K., H.A., C.G.S., R.R.R., J.H., J.D.C., V.A., J.N.N., I.P.L.J.M., O.K.P., J.M., L.R.K., A.K.M.). Data Analysis and Interpretation (C.C., F.R.T., H.G., J.R.O., K.L.K., K.M.N., L.C., M.P., M.B.S., M.E.D., S.D., S.K., Y.L., I.K., J.H.). Drafting the work or reviewing it critically for important intellectual content (All authors). Supervision/Mentorship (C.C., F.R.T., H.G., J.R.O., M.B.S., S.M.T., S.K.). Funding Acquisition (C.C., F.R.T., J.R.O., L.C., M.P., M.B.S.).

## Competing interests

A.F.: Employee of BlueWillow Biologics, Inc. at time of study design. C.G.S.: Institution has received research funds from BlueWillow Biologics, Inc. to support assays performed for this study. C.C.: Employee of BlueWillow Biologics, Inc. D.M.S.: Employee of BlueWillow Biologics, Inc. at time of study design. HA: Employee of BlueWillow Biologics, Inc. I.K.: Institution has received research funds from BlueWillow Biologics, Inc. to support assays performed for this study. I.P.L.J.M.: Institution has received research funds from BlueWillow Biologics, Inc. to support assays performed for this study. J.D.C.: Institution has received research funds from BlueWillow Biologics, Inc. to support assays performed for this study. J.H.: Institution has received research funds from BlueWillow Biologics, Inc. to support assays performed for this study. J.J.O.: Institution has received research funds from BlueWillow Biologics, Inc. to support assays performed for this study. J.N.N.: Institution has received research funds from BlueWillow Biologics, Inc. to support assays performed for this study. J.P.B.: Institution has received research funds from BlueWillow Biologics, Inc. to support assays performed for this study. J.R.O.: Personal: Advisory boards for GSK (RSV Cost Effectiveness Advisory Board, Vaccine Virtual Days 2023), Pfizer (Advisory Committee of Influenza Experts [ACTIvE]), and Moderna (New Vaccines Advisory Board). Consultant for ENA Respiratory. Institution has received research funds from BlueWillow Biologics, Inc. to support assays performed for this study. K.M.N.: Institution has received research funds from BlueWillow Biologics, Inc. to support assays performed for this study. K.L.K.: Institution has received research funds from BlueWillow Biologics, Inc. to support assays performed for this study. M.B.S.: Institution has received research funds from BlueWillow Biologics, Inc. to support assays performed for this study. M.E.D.: Institution has received research funds from BlueWillow Biologics, Inc. to support assays performed for this study. M.F.M.: Institution has received research funds from BlueWillow Biologics, Inc. to support assays performed for this study. M.P.: Institution has received research funds from BlueWillow Biologics, Inc. to support assays performed for this study. R.R.R.: Presently employed at Moderna Therapeutics and may own shares as a condition of employment. Completed involvement in this work prior to this employment. While at University of Maryland, her institution received research funds from BlueWillow Biologics, Inc. to support assays performed for this study. S.D.: Institution has received research funds from BlueWillow Biologics, Inc. to support assays performed for this study. S.M.T.: Institution has received research funds from BlueWillow Biologics, Inc. to support assays. performed for this study. T.H.: Employee of BlueWillow Biologics, Inc. at time of study design. V.A.: Institution has received research funds from BlueWillow Biologics, Inc. to support assays performed for this study. Y.L.: Institution has received research funds from BlueWillow Biologics, Inc. to support assays performed for this study. No other authors have competing interests.

## Additional information

[1]Center for Vaccine Development and Global Health, University of Maryland School of Medicine, Baltimore, MD, USA. [2]Division of Viral Products, Center for Biologics Evaluation and Research (CBER), US Food and Drug Administration, Silver Spring, MD, USA. [3]BlueWillow Biologics, Inc., Ann Arbor, MI, USA. [4]Department of Epidemiology and Public Health, University of Maryland School of Medicine, Baltimore, MD, USA. [5]Department of Microbiology and Immunology, University of Maryland School of Medicine, Baltimore, MD, USA. [6]These authors contributed equally: Meagan E. Deming, Franklin R. Toapanta. ✉e-mail: jortiz@som.umaryland.edu

## the rH5 Writing Group

Hugo Acosta[3], Vaidehi Agrawal[1], James D. Campbell[1], Jingping Hu[1], Ifayet P. L. Johnson-Mayo[1], Insung Kang[1], Lisa R. King[2], Jody Manischewitz[2], Ashish K. Mishra[2], Joseph N. Nkeze[1], Olivia Posadas[2], Rekha R. Rapaka[1], Cosette G. Schneider[5] & Douglas M. Smith[3]

