## [Peer Review file · Nature Communications]

An intranasal adjuvanted, recombinant influenza A/H5 vaccine primes against diverse H5N1 clades: a phase I trial

Corresponding Author: Professor Justin Ortiz

Version 0:

Reviewer comments:

Reviewer #2

(Remarks to the Author)

The authors have addressed my comments.

Reviewer #3

(Remarks to the Author)

I thank the authors for addressing my comments.

Regarding my original comment #1: I agree that without any measurable baseline titers against clade 2.1 or 1, there is no reason to assume that baseline titers against clade 2.2, clade 2.2.1, clade 2.3.4, and clade 2.3.4.4b were present. I was only commenting that the denominator for calculating %seroconversion should be 8 for each group, instead of 7,7,3, or 0, which includes only those with clade 2.2, 2.2.1, 2.3.4, or 2.3.4.4b measured.

REVIEWERS' COMMENTS

Reviewer #2 (Remarks to the Author):

The authors have addressed my comments.

Reviewer #3 (Remarks to the Author):

I thank the authors for addressing my comments.

Regarding my original comment #1: I agree that without any measurable baseline titers against clade 2.1 or 1, there is no reason to assume that baseline titers against clade 2.2, clade 2.2.1, clade 2.3.4, and clade 2.3.4.4b were present. I was only commenting that the denominator for calculating %seroconversion should be 8 for each group, instead of 7,7,3, or 0, which includes only those with clade 2.2, 2.2.1, 2.3.4, or 2.3.4.4b measured.

AUTHOR RESPONSE: WE HAVE REVISED THE MANUSCRIPT AND SUPPLEMENTAL TABLES THAT DESCRIBE SEROPROTECTION AGAINST CLADE 2.2, CLADE 2.2.1, CLADE 2.3.4, AND CLADE 2.3.4.4B PER THE PEER-REVIEWER SUGGESTION.